# Microglia-Derived Spp1 Promotes Pathological Retinal Neovascularization via Activating Endothelial Kit/Akt/mTOR Signaling

**DOI:** 10.3390/jpm13010146

**Published:** 2023-01-11

**Authors:** Qian Bai, Xin Wang, Hongxiang Yan, Lishi Wen, Ziyi Zhou, Yating Ye, Yutong Jing, Yali Niu, Liang Wang, Zifeng Zhang, Jingbo Su, Tianfang Chang, Guorui Dou, Yusheng Wang, Jiaxing Sun

**Affiliations:** 1Department of Ophthalmology, Xijing Hospital, Fourth Military Medical University, Xi’an 710032, China; 263750 Army Hospital of Chinese PLA, Xi’an 710043, China; 3Lintong Rehabilitation Center of PLA Joint Logistics Support Force, Xi’an 710600, China; 4College of Life Science, Northwestern University, Xi’an 710069, China; 5Department of Ophthalmology, The Northern Theater Air Force Hospital, Shenyang 110041, China; 6Eye Institute of Chinese PLA, Fourth Military Medical University, Xi’an 710032, China; 7Department of Neurobiology, School of Basic Medicine, Fourth Military Medical University, Xi’an 710032, China

**Keywords:** microglia, Spp1, endothelial cell, Kit, retinal neovascularization

## Abstract

Pathological retinal neovascularization (RNV) is the main character of ischemic ocular diseases, which causes severe visual impairments. Though retinal microglia are well acknowledged to play important roles in both physiological and pathological angiogenesis, the molecular mechanisms by which microglia communicates with endothelial cells (EC) remain unknown. In this study, using single-cell RNA sequencing, we revealed that the pro-inflammatory secreted protein Spp1 was the most upregulated gene in microglia in the mouse model of oxygen-induced retinopathy (OIR). Bioinformatic analysis showed that the expression of Spp1 in microglia was respectively regulated via nuclear factor-kappa B (NF-κB) and hypoxia-inducible factor 1α (HIF-1α) pathways, which was further confirmed through in vitro assays using BV2 microglia cell line. To mimic microglia-EC communication, the bEnd.3 endothelial cell line was cultured with conditional medium (CM) from BV2. We found that adding recombinant Spp1 to bEnd.3 as well as treating with hypoxic BV2 CM significantly enhanced EC proliferation and migration, while Spp1 neutralizing blocked those CM-induced effects. Moreover, RNA sequencing of BV2 CM-treated bEnd.3 revealed a significant downregulation of Kit, one of the type III tyrosine kinase receptors that plays a critical role in cell growth and activation. We further revealed that Spp1 increased phosphorylation and expression level of Akt/mTOR signaling cascade, which might account for its pro-angiogenic effects. Finally, we showed that intravitreal injection of Spp1 neutralizing antibody attenuated pathological RNV and improved visual function. Taken together, our work suggests that Spp1 mediates microglia-EC communication in RNV via activating endothelial Kit/Akt/mTOR signaling and is a potential target to treat ischemic ocular diseases.

## 1. Introduction

Pathological retinal neovascularization (RNV) is a defining feature of many ischemic ocular diseases, including retinopathy of prematurity (ROP), retinal vein occlusion (RVO), and diabetic retinopathy (DR), all of which result in irreversible vision loss [1]. Vascular endothelial growth factor (VEGF) has been proved to be the most powerful angiogenic mediators in RNV [2], and anti-VEGF drugs have transformed the treatment landscape for ocular angiogenic diseases. However, its therapeutic efficacy is generally limited, and the precise pathogenic mechanism of RNV remains largely unknown. Therefore, further demonstration of pathogenesis is critical for developing alternative therapeutic modalities.

As the resident immune cells in retina, microglia are required for shaping vascular development and are responsible for irregular vessel growth [3]. Accumulation of microglia have been reported to be a driving force of neovascular ocular diseases, including age-related macular degeneration (AMD) [4], DR [5], and Macular Telangiectasia (MacTel) [6]. During pathological RNV, activated microglia migrate to affected areas and promote vascular formation by up-regulating phagocytic activity and producing pro-inflammatory and pro-angiogenic mediators [3]. It has been reported that ablation of microglia could suppress RNV in oxygen-induced retinopathy (OIR), a mouse model of ischemic retinopathy [7]. Though several pathways, including Toll-like receptors [8], Basigin [9], inducible nitric oxide synthase 2 (Nos2, also known as iNOS) [10], and receptor-interacting serine-threonine kinase 3 (Ripk3, also known as RIP3) [11], have been reported to play roles in microglia-endothelial cell (EC) communication in RNV, the underlying molecular mechanisms remain unknown. Moreover, the pro-angiogenic effects of microglia does not rely on its physical contact with ECs, suggesting the critical roles of microglia-derived soluble factors [12].

Spp1 (also known as Osteopontin) is a secreted multifunctional glyco-phosphoprotein that responds to acute and chronic inflammatory settings and signals via integrin and CD44 receptors [13]. Spp1 is expressed in multiple cell types and has been proved to mediate a variety of cellular function such as cell growth [14], adhesion [14] and migration [15,16]. In central nervous system (CNS), microglia have been shown to be a major source of Spp1 under stress conditions, especially related to angiogenesis process [17]. However, the role and underlying regulatory mechanism of Spp1 in microglia, especially in microglia-EC communication during pathological RNV are not understood.

Kit (also known as c-KIT), one of the type III tyrosine kinase receptors, plays a critical role in numerous cellular processes, including cell survival, proliferation, growth and development, and neoplasia [18]. Kit binding to stem cell factor (SCF) results in protein kinase activation, including PI3-kinase (PI3K)/Akt/mammalian target of rapamycin (mTOR), phospholipase γ, mitogen-activated protein kinase (MAPK) and Src kinase pathways [19]. Kit activation enhances EC proliferation, survival, and migration, and therefore promotes angiogenesis [20,21]. In retina, it has been reported that Kit signaling is activated in ECs in murine models of pathological ocular neovascularization and promotes vessel growth [22]. Yet whether Kit participates in microglia-EC interaction remains unknown.

Here, we revealed that Spp1 was upregulated in pathological RNV and dominantly secreted by microglia using OIR model with single-cell RNA sequencing. Furthermore, we showed that microglia-derived Spp1 promoted EC proliferation and migration via endothelial Kit/Akt/mTOR activation. Finally, we investigated that intraocular application of Spp1 neutralizing antibody could suppress RNV and improve visual function. Our results underline the pivotal role of Spp1 in linking microglia activation and pathological RNV formation in ischemic retinopathy, which paves the way for novel therapeutic interventions.

## 2. Materials and Methods

### 2.1. Animals

C57/BL6 mice were maintained under specific pathogen-free conditions with a 12-h (h) light/dark cycle. All animal studies were approved by the Animal Experiment Administration Committee of the Fourth Military Medical University.

### 2.2. Mouse Model of OIR

The OIR model was induced as previously described [23]. At postnatal day 7 (P7), pups and their mothers were placed in a high-oxygen (75 ± 2%) chamber and returned to room air (normoxic) conditions at P12. Pups raised in room air at the same time were used as age-matched controls.

For in vivo Spp1 neutralization, OIR mice on P12 were intravitreally injected with 1 µL phosphate-buffered saline (PBS) containing 50 ng anti-Spp1 (AF808, R&D Systems, Minneapolis, MN, USA) in left eye and the same amount of IgG control antibody in the contralateral eye, using a 2.5-mL 34G Hamilton syringe (Hamilton, Reno, NV, USA). Pups were anesthetized and killed at P17.

For microglia removal, colony-stimulating factor 1 receptor (CSF1R) inhibitor PLX5622 (Selleck, Shanghai, China) was given daily with oral gavage to OIR mice from P12, as described before. Pups were anesthetized and killed at P17 [24].

### 2.3. Cell Culture and Treatment

The murine brain endothelial cell line bEnd.3 (ATCC) and murine microglial cell line BV-2 (ATCC) were cultured in Dulbecco’s modified eagle medium (DMEM) (HyClone, Logan, UT, USA) with 10% fetal bovine serum (FBS, HyClone), 100 units/mL penicillin G, and 100 μg/mL streptomycin (Gibco, Thermo Fisher Scientific, Waltham, MA, USA). Cells were cultured at 37 °C with 5% CO_2_ and air for normoxic treatment, and with 5% CO_2_ and 1% O_2_ for hypoxia.

LPS (MedChemExpress, South Brunswick Township, NJ, USA) at 1 μg/mL was added to stimulate BV2 cells for 24 h. For blockade studies, inhibitors, including NF-κB inhibitor BAY 11-7082 (10 µM, MedChemExpress) and HIF-1α inhibitor GN44028 (10 µM, MedChemExpress), were added to BV2 cell culture 1 h before stimulation.

To study microglia-EC communications in vitro, bEnd.3 culture media were replaced with the CM of BV2 cells cultured under normoxia (normoxia MG-CM) or hypoxia (hypoxia MG-CM), and then treated with 50 ng/μL anti-Spp1 (AF808, R&D Systems) or IgG control antibody if needed, for 2 h at 37 °C. In a separate study, recombinant mouse Spp1 protein (rSpp1, 0.1/0.5/1.0 μg/mL, 441-OP, R&D Systems) was added to directly stimulate bEnd.3 cells. Kit inhibitor Imatinib Mesylate (10 µM, MedChemExpress) was added to into bEnd.3 after stimulation with MG-CM to block Kit activation.

### 2.4. Immunostaining

For vascular analysis, retinal tissues were flat mounted at P12 or P17 and examined under a confocal laser scanning microscopy (FV1000, Olympus) as described before [25]. After being post-fixed in 4% paraformaldehyde (PFA), the retinal tissues were blocked and permeabilized overnight in blocking buffer (PBS containing 1% bull serum albumin (BSA) and 0.5% Triton X-100). Primary antibodies, including anti-Iba1 antibody (1:500, 019-19741, Wako) and anti-Spp1 antibody (1:20, AF808, R&D Systems, Minneapolis, MN, USA), were diluted in blocking buffer and incubated overnight at 4 °C. After three PBS washes, Fluorescein-labeled Griffonia Simplicifolia Lectin I (GSL I) isolectin B4 (1:100, FL-1201, Vector Labs, Burlingame, USA) and following secondary antibodies, including Alexa Fluor 594-conjugated donkey anti-goat IgG (A-11058, Invitrogen, Waltham, MA, USA) and Alexa Fluor 647-conjugated donkey anti-rabbit IgG (A-31573, Invitrogen), were incubated subsequently overnight at 4 °C in PBS. The retinal tissues were then washed in PBS and flat-mounted on glass slides for photographing.

For frozen section analysis, eyeballs were denucleated and fixed in 4% PFA and dehydration in 30% sucrose in PBS overnight. Tissues were embedded in optimal cutting temperature (OCT) compound (Sakura Finetek, Torrance, CA, USA) and cryosectioned at 10 μM thickness. Frozen sections were dried for 2 h at room temperature, blocked, permeabilized, and incubated with primary antibodies, including anti-Iba1 antibody (1:500, 019-19741, Wako, Osaka, Japan), anti-NeuN antibody (1:500, ab104224, Abcam, Cambridge, UK), anti-Slc1a3 antibody (1:200, 20785-1-AP, ProteinTech Group, Rosemont, IL, USA), anti-Pecam1 antibody (1:100, 102502, BioLegend, San Diego, CA, USA) and anti-Spp1 antibody (1:20, AF808, R&D Systems) at 4 °C overnight. After washing, sections were incubated with secondary antibodies for 2 h at room temperature. DAPI staining was performed according to the manufacturer’s instructions. Images were captured under a laser-scanning confocal fluorescence microscope (FV1000, Olympus, Tokyo, Japan).

### 2.5. RNA Sequencing Analysis

#### 2.5.1. Single-Cell RNA Sequencing

The single-cell RNA sequencing data used in this study have been deposited in the National Center for Biotechnology Information’s (NCBI) Sequence Read Archive (SRA) database (PRJNA864092, https://www.ncbi.nlm.nih.gov/sra (accessed on 1 August 2022)). Bioinformatic analysis was performed using Omicsmart (http://www.omicsmart.com (accessed on 24 September 2020)), an online platform for data analysis.

#### 2.5.2. Bulk RNA Sequencing

bEnd.3 cells were stimulated with hypoxia MG-CM and treated with anti-Spp1 or IgG control for 24 h. RNA extraction and sequencing were performed using Illumina Novaseq 6000 by Gene Denovo Biotechnology Co., Ltd. (Guangzhou, China). Raw data of RNA sequencing have been uploaded in the Genome Sequence Archive of the BIG Data Center (CRA008550, http://bigd.big.ac.cn/gsa (accessed on 19 October 2022)). Bioinformatics analysis was performed using Omicsmart (http://www.omicsmart.com (accessed on 7 January 2022)).

### 2.6. Quantitative Reverse Transcription Polymerase Chain Reaction (qRT-PCR)

Tissues and cells were lysed in TRIzol reagent (Thermo Fisher, Waltham, MA, USA) in accordance with the manufacturer’s instructions. RNA was isolated, purified and then reverse transcribed to cDNA using PrimeScript^™^ RT Master Mix (Takara, Kusatsu, Japan). qRT-PCR was performed as previously mentioned with the primer sequences shown in Table 1 [26]. Amplification of β-actin was used as an internal control. The levels of gene expression were reported as relative fold change compared to control group.

### 2.7. Western Blotting

Cells or tissues were lysed in protease inhibitor cocktail (1697498, Roche)-added radio immunoprecipitation assay (RIPA) buffer (Beyotime, Nanjing, China). Sodium dodecyl sulfate–polyacrylamide gel electrophoresis (SDS-PAGE) was used for protein separation. The primary antibodies used for immunoblotting on polyvinylidene fluoride (PVDF) membranes were as follow: anti-Spp1 antibody (1:1000, ab8448, Abcam), anti-HIF-1α antibody (1:500, WL01607, Wanleibio, Shenyang, China), anti-NF-κB p65 antibody (1:1000, 8242, Cell Signaling Technology, Danvers, MA, USA), anti-Phospho-NF-κB p65 antibody (1:1000, 3033, Cell Signaling Technology), anti-Kit antibody (1:1000, A0357, Abclonal, Woburn, MA, USA) and anti-GAPDH antibody (1:10,000, 10494-1-AP, ProteinTech Group). Secondary antibodies containing horseradish peroxidase (HRP)-linked goat anti-rabbit IgG and horse anti-mouse IgG antibody (1:2000, 7074 and 7076, Cell Signaling Technology) were incubated after three washes in PBS with 0.1% Tween-20 (PBST). The membranes were scanned using an enhanced chemiluminescence (ECM) assay (Beijing 4A Biotech Co., Ltd., Beijing, China). Analysis of the protein bands were performed using ImageJ software (version 1.49p; National Institutes of Health).

### 2.8. Cell Proliferation Assay

EdU (5-ethynyl-2′-deoxyuridine) (C10310-1, RiboBio, Guangzhou, China) incorporation was performed according to manufacturer’s instructions as described previously [27]. In brief, bEnd.3 were incubated with 50 μM EdU for 2 h and then fixed with 4% PFA for 30 min (min). Cells were then stained with Apollo 567 and Hoechst. Images were taken using a confocal laser scanning microscopy (FV1000, Olympus) and analyzed using Image-Pro Plus 6.0 (Media Cybernetics, Rockville, MD, USA).

### 2.9. Cell Migration Assay

Wound healing assay was used to test cell migration ability as described before [28]. In brief, after making a scratch with a p20 pipette tip, images were taken at 0/12/24/48 h to monitor the rate of wound closure under a microscope (CKX41, Olympus) with a CCD camera (DP70, Olympus) and analyzed using Image-Pro Plus 6.0 (Media Cybernetics).

### 2.10. OIR Lesion Assessment

OIR lesion evaluation was performed in accordance with the published protocol [29]. Activated microglia were recognized by positive immunostaining with Iba1 with short and thick processes [30]. The number of Spp1^+^ were determined at P12 and P17. Three representative fields in the central (avascular) and peripheral (neovascular, NV) zone were chosen in each retinal flat mount for quantification. The mean number of Spp1 from three fields of the same retina was calculated as the average value. Six retinas were calculated for each group. Areas of NV tufts and the whole flat-mount retina were outlined using a manual masked observer for quantification. Three pups were examined for each group.

### 2.11. ERG Analysis

ERG responses were recorded in accordance with the International Society for Clinical Electrophysiology of Vision’s recommendations as described previously [25]. In brief, mice were dark adapted overnight, anesthetized using 1% sodium pentobarbital (P3761, Sigma, St. Louis, MO, USA) and sumianxin II (Jilin Shengda, Jilin, China) under dim red-light conditions, with pupils dilated. The active electrode was attached to the center of the cornea with a ring electrode, while the reference and the ground electrode were inserted beneath the skin of the cheek around the tested eye and tail, respectively. The full-field (Ganzfeld) stimulation and a computer system (RETI port; Roland, Rüsselsheim, Germany) were used for recording responses.

### 2.12. Statistics

Statistical analysis and visualization were performed using GraphPad Prism 8.0 (GraphPad Software). All quantitative data were displayed as mean ± SD. The unpaired or paired Student’s *t* test, or one-way analysis of variance (ANOVA) were used to determine statistically significant difference between two or more groups. Significant difference was defined as *p* < 0.05.

## 3. Results

### 3.1. Spp1 Is Characteristically Upregulated in Microglia in OIR

To elucidate the molecular characteristics of microglia in hypoxia-induced angiogenesis, retinal tissues of OIR and control mice were collected and digested into single cell suspension for single-cell RNA profiling. All cells were grouped into 20 distinct clusters using the Seurat package, and then further defined as 12 types according to cell surface markers (Figure 1A). The clusters of microglia, with high expression of C-X3-C motif chemokine receptor 1 (Cx3cr1), transmembrane protein 119 (Tmem119), and other microglia signature genes, were selected for downstream analysis (Figure 1B). Differential gene expression analysis revealed 620 up-regulated genes and 548 down-regulated genes in microglia in OIR compared to the control, and most up-regulated genes were secretary cytokines and receptors, including Spp1, C-C motif chemokine ligand 2 (Ccl2), insulin-like growth factor 1 (Igf1), etc. (Figure 1C). Among those genes, Spp1 showed the highest fold change, and was specifically enriched in microglia in OIR retina (Figure 1C,D).

We then tested the Spp1 expression profile in vitro. The protein level of Spp1, as shown in Western blotting analysis, was higher in OIR mice than in the control at both postnatal day 12 (P12) and P17 (Figure 2A). Whole flat-mount and fluorescent staining analysis revealed that the number of Spp1 positive (Spp1^+^) cells increased significantly both in the central zone (zone with avascular area at P12 and NV tufts at P17) and peripheral zone (zone with mainly close normal vessels) in OIR retina at P12 and P17, respectively (Figure 2B,C). Moreover, Spp1^+^ cells were found in higher numbers around NV mass between the avascular central zone and the peripheral vascularized zone at OIR P17 (Figure 2C). Immunofluorescence further showed that Spp1 was mainly detected in extracellular spaces besides Iba1-possitive microglia surrounded vessels, and not fully co-expressed with microglia, retinal ganglion cells, Müller cells or ECs, suggested its role as secretory protein (Figure 2D,E and Appendix A). To further reveal the origin of Spp1, CSF1R inhibitor PLX5622 was used to clear microglia in OIR retina. Immunofluorescence showed that expression of Spp1 was significantly decreased (Appendix A) and only appeared in areas with residual microglia (Appendix A) after PLX5622 treatment. These results demonstrate that Spp1 is upregulated in OIR mice and mainly secreted from the microglia in response to RNV.

### 3.2. Expression of Spp1 Is Regulated by HYPOXIA-Inducible Factor 1 (HIF-1) Signaling Pathway and NF-Kappa B (NF-κB) Signaling Pathway

To gain mechanistic insights underlying Spp1^+^ microglia in OIR, we figured out Spp1^+^ and Spp1-negative (Spp1^−^) microglia subclusters and compared gene expression profiles among the two subclusters. Differential gene expression and enrichment analysis revealed that the altered genes were significantly enriched in HIF-1 signaling pathway and NF-κB signaling pathway (Figure 3A). As hypoxia and inflammation are the main pathological environment in OIR which active microglia [9,29,31] (Appendix A), we hypothesized that upregulation of microglial Spp1 in OIR might due to HIF-1-mediated hypoxic and NF-κB-mediated inflammatory stimuli. In order to verify this hypothesis, we treated BV2 cells with hypoxia (1% O_2_) or Lipopolysaccharide (LPS). We found that expression of Spp1 and HIF-1α were increased in microglia cultured under hypoxic condition and could be inhibited by pre-treatment with HIF-1α inhibitor GN44028 (Figure 3B). Moreover, LPS treatment could significantly upregulate both the expression and phosphorylation level of NF-κB p65 and the expression of Spp1, which could be abolished by NF-κB inhibitor BAY-77028 (Figure 3C). These findings suggest that microglia-derived Spp1 might be regulated downstream to hypoxia and inflammation via activation of HIF-1 and NF-κB signaling pathway, respectively.

### 3.3. Spp1 Mediates Microglia-EC Interaction and Promotes EC Proliferation and Migration In Vitro

We next investigated the role of secreted Spp1 in regulating endothelial function. Adding rSpp1 could significantly increase EC proliferation and migration (Figure 4A,B). To further test whether Spp1 played a role in microglia-EC communication, cultured bEnd.3 cells were treated with DMEM or conditional medium (CM) from microglia (MG-CM), respectively. We found that CM from hypoxia-treated microglia promoted EC proliferation and motility capacity, compared with DMEM or normoxia-treated CM (Appendix A). Moreover, adding an Spp1-neutralizing antibody (anti-Spp1) significantly repressed those effects (Figure 4C,D). Taken together, these data indicate that microglia-derived Spp1 promotes EC proliferation and migration.

### 3.4. Spp1 Activates Kit/Akt/mTOR Signaling Cascade in EC

For further understanding the underlying mechanisms of Spp1 in promoting EC proliferation, bulk RNA-seq was performed to compare transcriptomes of bEnd.3 treated with hypoxia MG-CM plus anti-Spp1 or IgG control. We found that Kit, a well-known cell proliferation-related gene, was significantly downregulated in anti-Spp1-treated group, which was further tested by qRT-PCR and Western blotting (Figure 5A–C). Moreover, anti-Spp1 treatment suppressed expression and phosphorylation level of Akt and mTOR (Figure 5D). To confirm the function of Spp1 in attenuating the Kit/Akt/mTOR pathway, we simultaneously treated bEnd.3 with rSpp1, imatinib mesylate (a Kit inhibitor), and the combination of both. Results from Western blotting showed that Spp1 overexpression significantly increased Kit expression as well as the expression and phosphorylation level of Akt/mTOR pathway, which could be reduced by imatinib mesylate treatment (Figure 5E,F). These findings suggest that Spp1 promotes cell proliferation likely by activating Kit signaling cascade.

### 3.5. Interference of Spp1 Inhibits RNV and Alleviate Visual Injury in OIR Model

At last, we assessed the effect of Spp1 on retinal neovascular formation and visual function by intravitreal injection of anti-Spp1 in vivo. IgG control or anti-Spp1 antibody were injected intravitreally at P12 of OIR mice, and whole retina tissues were collected at P17 (Figure 6A). As expected, Spp1 protein levels were significantly lower in anti-Spp1 antibody-injected retinas than in control groups (Figure 6B). Mice treated with anti-Spp1 antibody showed a significantly reduced area of NV tufts compared with the control group (Figure 6C). We next measured ERG to examine visual function. b-wave amplitudes of the Dark-adapted 0.01 ERG and Light-adapted 3.0 ERG, the a- and b-wave amplitudes of Dark-adapted 3.0 ERG, and P3 amplitudes of the Dark-adapted 3.0 Ops response were compared in normal control and IgG control or anti-Spp1 antibody-injected eyes. It was observed that the anti-Spp1 antibody improved ERG amplitude, indicating a better neural response (Figure 6D). Consequently, this data confirms that intraocular application of the anti-Spp1 antibody could suppress retinal neovascular formation and alleviate visual injury in vivo, providing evidence for Spp1 as a potential therapeutic target.

## 4. Discussion

Results of this study showed that: (a) Spp1 was mainly secreted from active microglia in pathological RNV; (b) expression of Spp1 was regulated by HIF-1 signaling pathway and NF-κB signaling pathway; (c) microglia-derived Spp1 enhanced EC proliferation and migration via activating Kit/Akt/mTOR signaling cascade; and (d) intravitreal injection of anti-Spp1 suppressed retinal neovascular formation and alleviate visual injury in OIR. Taken together, our data suggested that Spp1 mediated microglia-EC communication and promoted RNV, which might provide a potential therapeutic target.

Pathological neovascularization, manifested by the irregular and rapid proliferation of ECs from vascular clusters, is a major feature of ischemic retinopathies. Recent studies have shown that retinal microglia accumulation is closely associated with formation of NV tufts or fibrovascular membrane as well as retinal degeneration in patients with AMD or DR [4,5,6], providing new hope for microglia-targeting therapies. As microglia are major resident immune cells in CNS and play a key role in regulating vasculature, more concern is given to how microglia communicate with ECs in pathological angiogenesis. Therefore, it is necessary to better understand distinct expression characters of microglia in ischemic retinopathy. Here, we revealed, as shown by single-cell RNA sequencing analysis, that the inflammatory cytokine Spp1 was the most influenced gene upregulated in microglia in mouse model of OIR. Moreover, in accordance with previous reports that activated microglia and macrophages are the main source of Spp1 in CNS [32,33], we found that Spp1 mRNA was mainly elevated in microglia and then macrophages. Further immunofluorescent staining showed that Spp1 protein was distributed within microglia or around perivascular microglia, suggesting that it is mainly synthesized by microglia but acts in a secreted form in neovascular ECs. Moreover, Spp1^+^ microglia showed an altered gene expression profile in the HIF-1α and NF-κB signaling pathway, indicated a response to hypoxia and inflammation. As hypoxia and inflammation have been confirmed to play important roles in OIR pathology [29,34], we further confirmed an upregulation of Spp1 in BV2 responded to those two stimuli, which could be blocked by respectively inhibiting HIF-1α and NF-κB pathway. Therefore, we concluded that the hypoxic and inflammatory condition of OIR might induce microglial Spp1 upregulation via the HIF-1α and NF-κB pathway.

Spp1 is reported to be involved in angiogenesis, including post-myocardial ischemic neovascularization [35,36], peripheral artery disease and diabetes mellitus [37,38], and AMD [39]. Though most reports indicated that Spp1 play a pro-angiogenic role [40,41,42,43], research also showed an anti-angiogenic role of Spp1 [39]. These controversial effects might be attributed to different treat strategies or target cells. Here, we assumed that microglial secreted Spp1 might regulate EC functions. Previous evidence has showed that glioma cell-derived Spp1 promoted endothelial progenitor cell proliferation, migration, and tube formation [44]. Yao et al. recently demonstrated that macrophages and microglia-derived Spp1 could regulate ECs through Itga5 and Itgb1 receptors in the lesion after spinal cord injury [45]. Here, using bEnd.3 cell line, we confirmed that Spp1 promoted EC proliferation and migration. Moreover, as shown by previous reports [46,47,48] and our results, CM from hypoxic cultured microglia promoted EC angiogenic ability. We found that an Spp1 blockade could reverse the pro-angiogenic effect of hypoxic microglia, suggesting secreted Spp1 functioned as a mediator regulating microglia-EC communication in pathological RNV and promoted angiogenesis.

Though it is known that Spp1 mainly binds to CD44 and integrins and regulate several kinase pathways [49], the downstream targets of Spp1 are barely studied in ECs. To further reveal the function of microglia-derived Spp1 on ECs, we added anti-Spp1 to hypoxia MG-CM treated bEnd.3 and performed RNA sequence. We found that Kit signaling cascades was blocked in anti-Spp1 treated ECs and might account for the phenotype changes.

Kit, a well-known type III receptor tyrosine kinases, is essential for numerous cellular processes [50]. Abnormal Kit activation is associated with a variety of tumorigenesis in humans [51]. Kit is expressed in mature ECs and promotes cell survival, migration, and tube formation [21]. Recent research has revealed that endothelial Kit activity is essential for angiogenic-dependent effective tissue repair [52,53] as well as pathological neovascularization [22]. Activation of Kit depends on binding of its physiological ligand SCF, after which receptor homodimerization occurs and results in autophosphorylation at tyrosine residues of the dimer, leading to subsequent protein kinase signal transduction pathways activation [50]. Among those well-established downstream signaling routes, the PI3K/Akt pathway plays a pivotal role on increased cell viability and proliferation. Moreover, the serine/threonine kinase mTOR complex 1 (mTORC1) is located downstream of Akt, which activation promotes S6-kinase-dependent translation [54]. It has been demonstrated that Kit signaling activate mTOR and S6-kinase in an Akt-dependent manner [55,56]. In the current study, we found that anti-Spp1 treatment suppressed both the expression and phosphorylation level of Akt and mTOR induced by hypoxia MG-CM stimulus. Moreover, rSpp1 treatment significantly increased Kit expression as well as expression and phosphorylation level of Akt/mTOR pathway, which could be partially rescued by Kit inhibition. Taken together, these findings suggest that microglia-derived Spp1 regulates EC proliferation via Kit/Akt/mTOR cascade.

Previous studies have shown that systemically therapeutic neutralization using an anti-Spp1 antibody could suppress lesion size in mice models of ischemic brain and eye diseases [40,41,57]. In this study, we used a more targeted approach that administers anti-Spp1 locally via intravitreal injection to restrict its side-effect on peripheral immune compartments. We found that local Spp1 suppression could successfully reduce neovascular formation and alleviate visual injury in OIR, which indicated its potential as a novel treatment target.

## 5. Conclusions

In summary, our data suggest that increased Spp1 secretion, possibly resulted from hypoxia and inflammation, represents a predominant change of microglia in pathological RNV. Microglia-derived Spp1 mediates microglia-EC communication partially via promoting Kit/Akt/mTOR signaling pathway-mediated EC growth and migration, and therefore aggravates abnormal vessel formation and subsequent functional injury. Therapeutic effects of anti-Spp1 antibody further indicate its potential as a novel target for treatment of ischemic retinopathies.

## Figures and Tables

**Figure 1 jpm-13-00146-f001:**
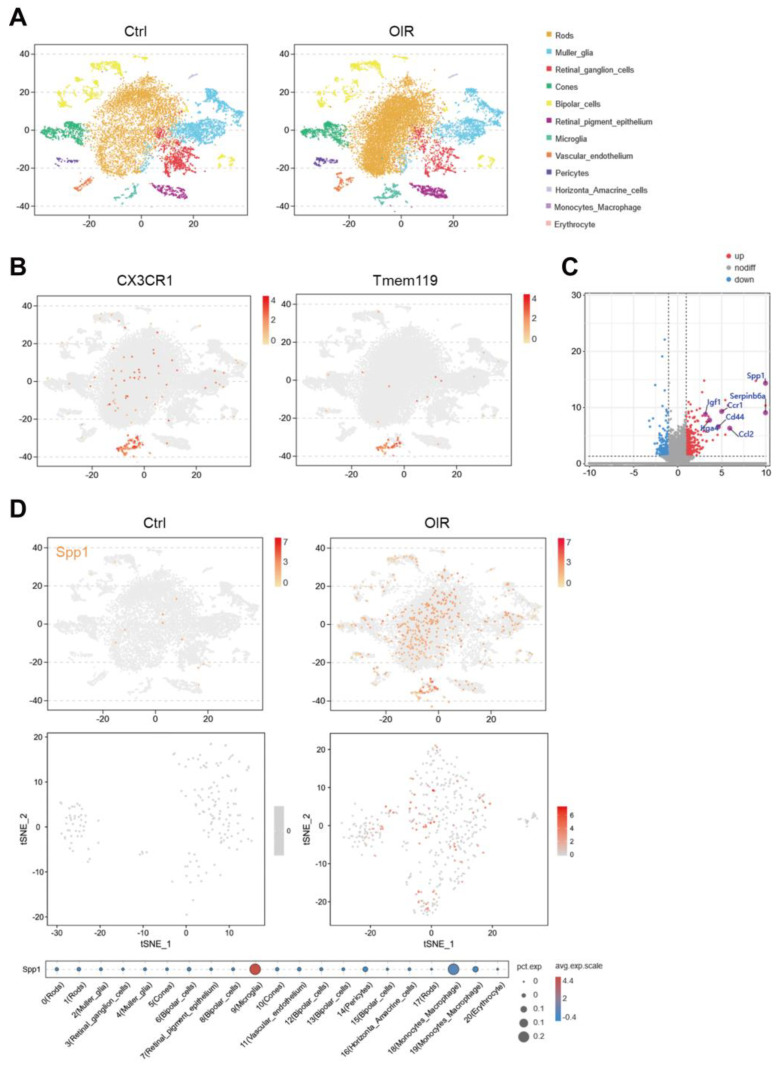
Single-cell RNA sequencing uncovers a predominant increased expression of Spp1 in microglia in OIR. (**A**) t-SNE plots of all single cells annotated by cell types from Ctrl and OIR. Different cell clusters were color-coded. (**B**) t-SNE maps showing a well-defined high expression of Cx3cr1 and Tmem119 in the cluster of microglia. (**C**) Volcano plot showing the significance and fold change of genes between microglia in Ctrl and OIR. Red dots represented up-regulated genes, while blue dots represented down-regulated genes. Several most-changed secretary cytokines and receptors are labeled, including Spp1. (**D**) t-SNE maps showing expression of Spp1 in all cells (**superior**) and microglia (**middle**). Bubble chart (**inferior**) showing that Spp1 expressed predominantly in microglia.

**Figure 2 jpm-13-00146-f002:**
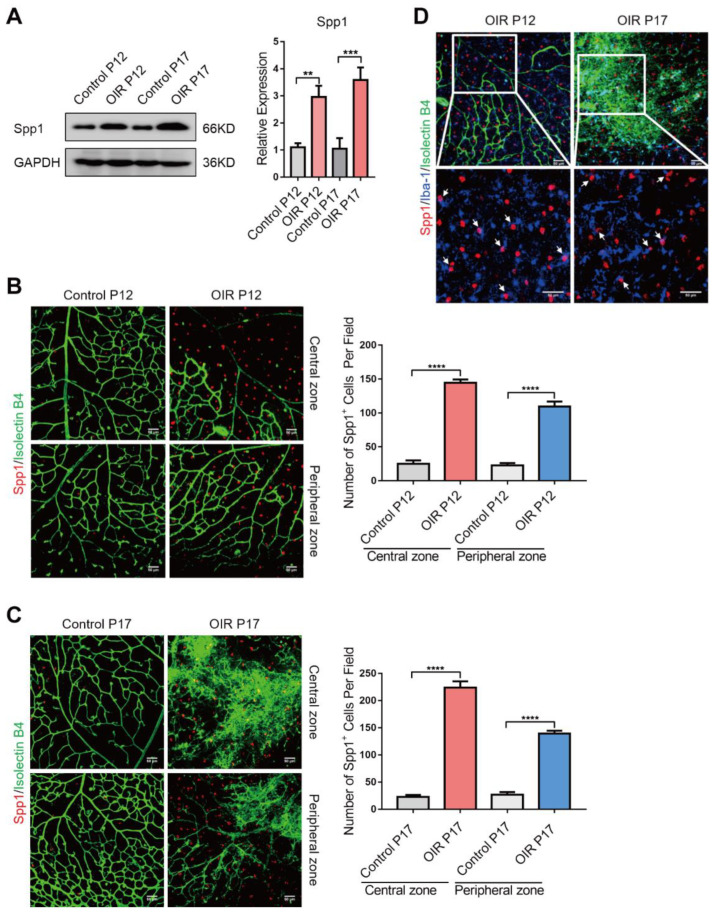
Spp1 is upregulated both in avascular zone and NV tufts in OIR. (**A**) Western blotting results showing that the expression levels of Spp1 protein were higher in OIR retinas than control at both P12 and P17. (mean ± SD; *n* = 3/group; one-way ANOVA). (**B**) Immunofluorescence images of retinal flat mounts in Control P12 and OIR P12 stained with IB4 (green) and anti-Spp1 (red). Numbers of Spp1^+^ cells were quantified separately at central zone and peripheral zone. (mean ± SD; *n* = 6/group, four visual fields were taken from one retina to calculate the average value; one-way ANOVA). Scale bars: 50 μm. (**C**) Immunofluorescence images of retinal flat mounts in Control P17 and OIR P17 stained with IB4 (green) and anti-Spp1 (red). Numbers of Spp1^+^ cells were quantified separately at central zone and peripheral zone. (mean ± SD; *n* = 6/group, four visual fields were taken from one retina to calculate the average value; one-way ANOVA). Scale bars: 50 μm. (**D**) Immunofluorescence images of retinal flat-mounts in OIR P12 and OIR P17 stained with IB4 (green), anti-Iba-1(blue) and anti-Spp1 (red), showing that the immunoreactivity of Spp1 was mainly co-labeled with or surrounding Iba-1. Scale bars: 50 μm. ** *p* < 0.01, *** *p* < 0.001, **** *p* < 0.0001.

**Figure 3 jpm-13-00146-f003:**
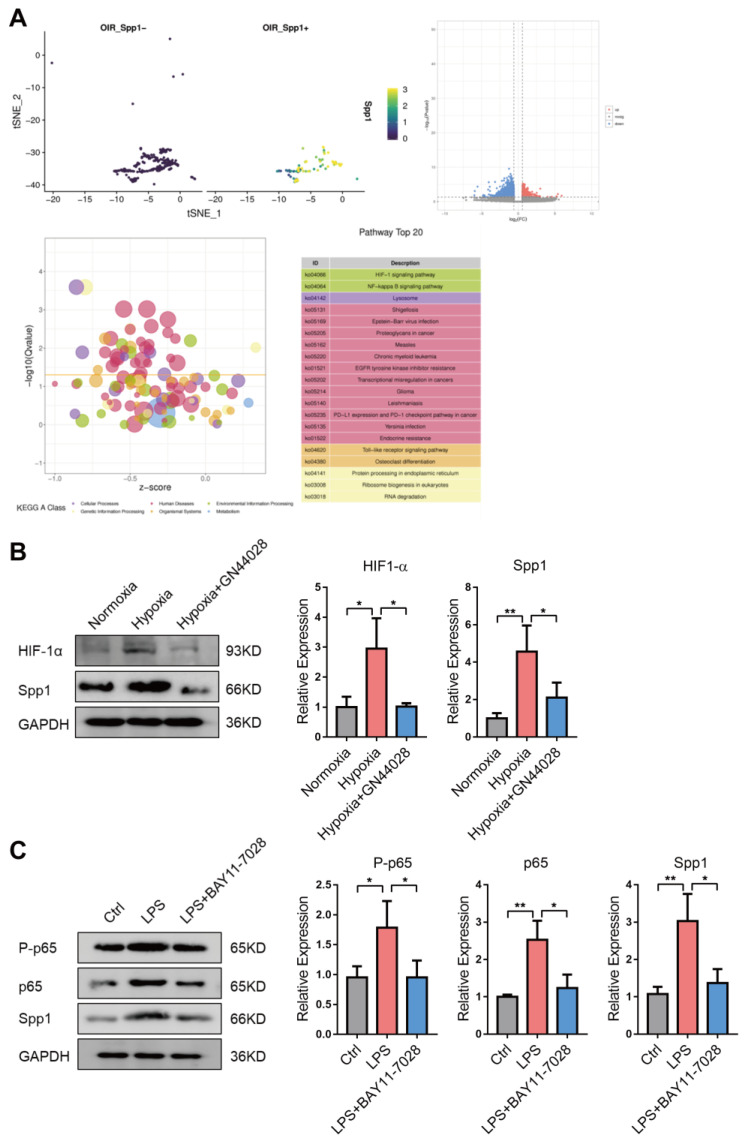
Expression of Spp1 in microglia is regulated by HIF-1 signaling pathway and NF-κB signaling pathway. (**A**) t-SNE plots depicting the separation of Spp1^−^ and Spp1^+^ subclusters of microglial in OIR. Volcano plot showing the significance and fold change of genes between Spp1^−^ and Spp1^+^ microglia. Red dots represented up-regulated genes, while blue dots represented down-regulated genes. Enrichment analysis showing a significant enrichment of different genes on HIF-1 signaling pathway and NF-κB signaling pathway. (**B**) Western blotting showing protein levels of HIF-1α and Spp1 in BV2 cells cultured under normoxia, hypoxia, and hypoxia plus HIF-1α inhibitor GN44028 (mean ± SD; *n* = 3/group; one-way ANOVA). (**C**) Western blotting showing protein levels of Spp1, p65 and P-p65 in BV2 cells treated with normal medium, LPS, and LPS plus NF-κB inhibitor BAY-77028 (mean ± SD; *n* = 3/group; one-way ANOVA). * *p* < 0.05, ** *p* < 0.01.

**Figure 4 jpm-13-00146-f004:**
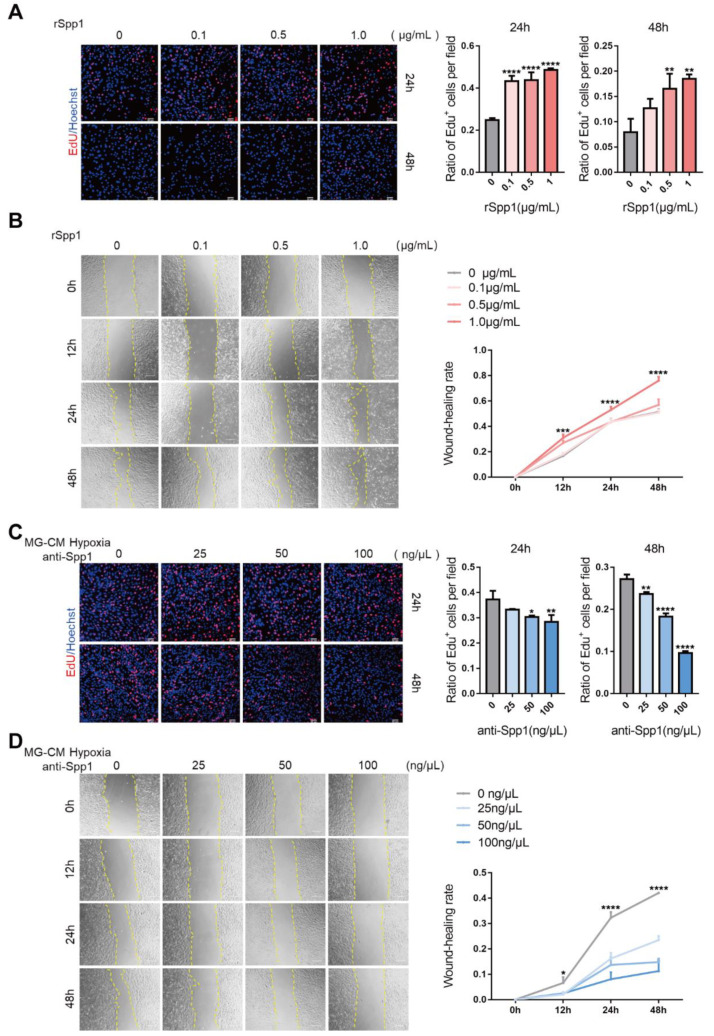
Stimulation with rSpp1 promotes EC proliferation and migration in vitro, while neutralization of Spp1 represses these effects induced by hypoxic MG-CM. (**A**) Results from EdU incorporation assay showing the proliferation capacity of bEnd.3 treated with rSpp1 of different concentrations (mean ± SD; *n* = 3/group, three visual fields were taken from one well to calculate the average value; one-way ANOVA). Scale bars: 50 μm. (**B**) Results from Wound healing assay showing the migration rate of bEnd.3 treated with rSpp1 of different concentrations (mean ± SD; *n* = 3/group, three visual fields were taken from one well to calculate the average value; one-way ANOVA). Scale bars: 200 μm. (**C**) Results from EdU incorporation assay showing the proliferation capacity of bEnd.3 treated with hypoxic MG-CM plus anti-Spp1 of different concentrations (mean ± SD; *n* = 3/group, three visual fields were taken from one well to calculate the average value; one-way ANOVA). Scale bars: 50 μm. (**D**) Results from Wound healing assay showing the migration rate of bEnd.3 treated with hypoxic MG-CM plus anti-Spp1 of different concentrations (mean ± SD; *n* = 3/group, three visual fields were taken from one well to calculate the average value; one-way ANOVA). Scale bars: 200 μm. * *p* < 0.05, ** *p* < 0.01, *** *p* < 0.0001, **** *p* < 0.0001.

**Figure 5 jpm-13-00146-f005:**
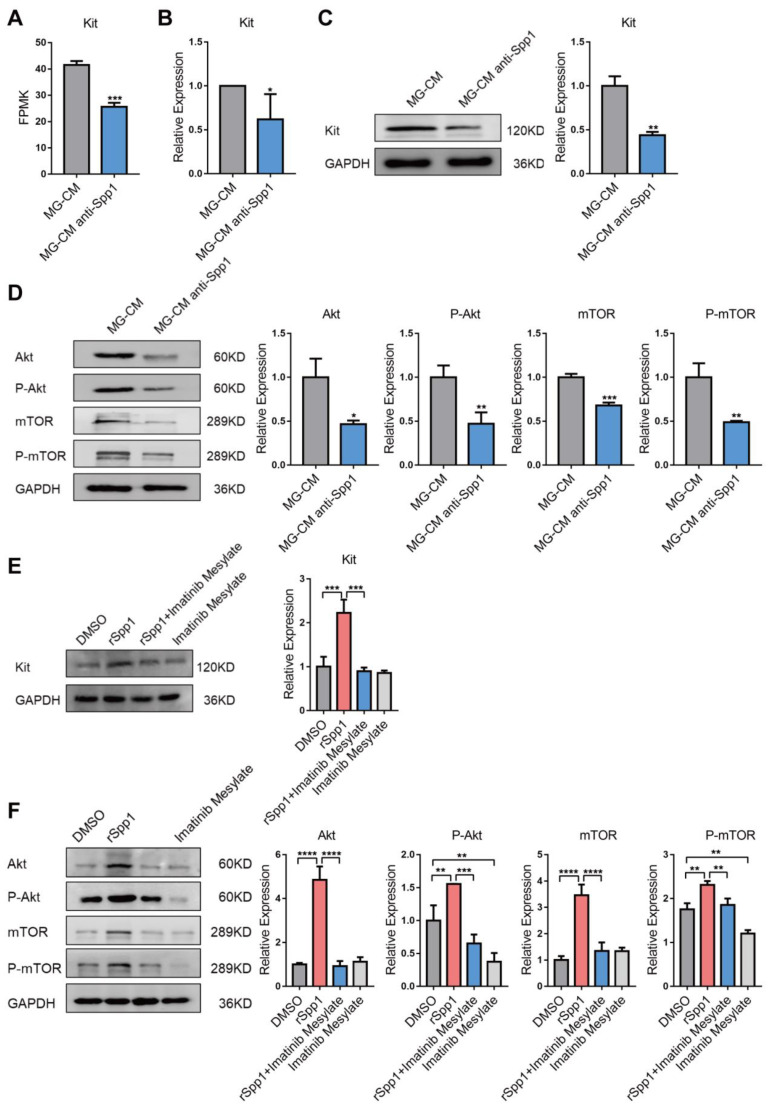
Microglia-derived Spp1 activating Kit-Akt-mTOR signaling in ECs. (**A**) RNA sequencing data revealing decreased fragments kilobase million (FPKM) of Kit in bEnd.3 treated with hypoxic MG-CM plus anti-Spp1 compared to those treated with hypoxic MG-CM only (mean ± SD; *n* = 3/group; unpaired Student’s *t*-test). (**B**) qRT-PCR results showing reduced expression level of Kit mRNA in anti-Spp1 treated group (mean ± SD; *n* = 3/group; paired Student’s *t*-test). (**C**) Western blotting analysis showing reduced expression level of Kit protein in anti-Spp1 treated group (mean ± SD; *n* = 3/group; unpaired Student’s *t*-test). (**D**) Western blotting analysis showing the protein level of Akt, P-Akt, mTOR and P-mTOR in bEnd.3 treated with hypoxic MG-CM or hypoxic MG-CM plus anti-Spp1 (mean ± SD; *n* = 3/group; unpaired Student’s *t*-test). (**E**) Western blotting analysis showing the protein level of Kit in bEnd.3 treated with DMSO, rSpp1, imatinib mesylate, and a combination of both (mean ± SD; *n* = 3/group; one-way ANOVA). (**F**) Western blotting analysis showing the protein level of Akt, P-Akt, mTOR and P-mTOR in bEnd.3 treated with DMSO, rSpp1, imatinib mesylate, and a combination of both (mean ± SD; *n* = 3/group; one-way ANOVA). * *p* < 0.05, ** *p* < 0.01, *** *p* < 0.0001, **** *p* < 0.0001.

**Figure 6 jpm-13-00146-f006:**
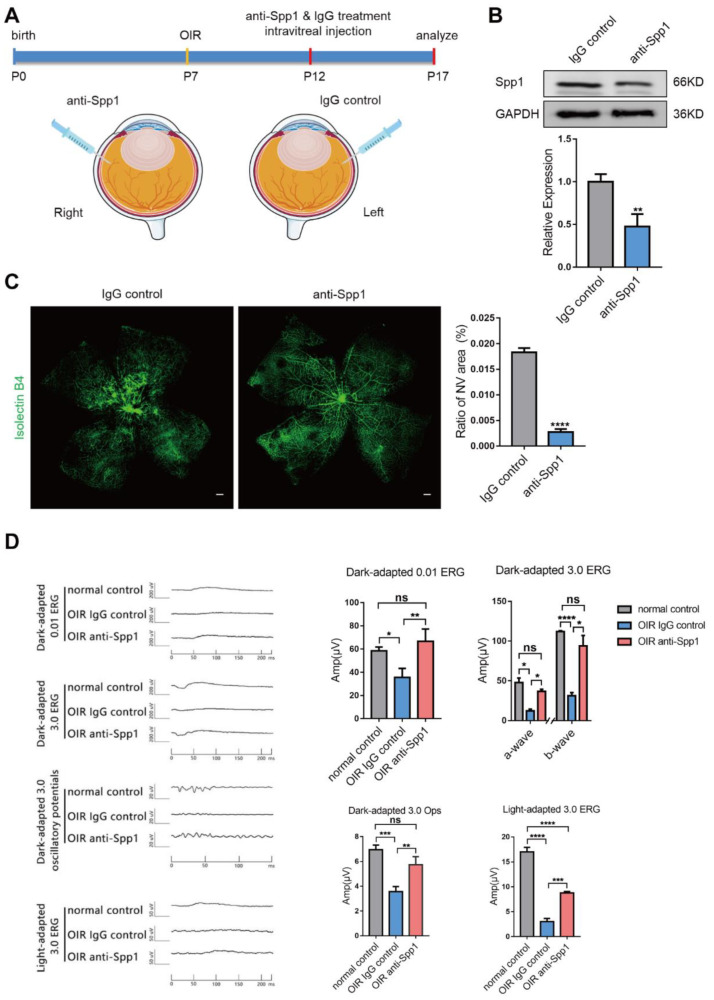
Interference of Spp1 on protein level inhibits retinal neovascular formation and improve visual function in OIR mice. (**A**) Experimental scheme. OIR mice were induced at P7. Pups were intravitreally injected with 1 µL PBS containing 50 ng anti-Spp1 in left eye and the same amount of IgG control antibody in the contralateral eye at P12 and sacrificed at P17 for analysis. (**B**) Western blotting analysis showing reduced expression level of Spp1 protein in anti-Spp1-injected group (mean ± SD; *n* = 3/group; unpaired Student’s *t*-test). (**C**) Immunofluorescence images of whole retinal flat-mounts in OIR P17 stained with IB4 (green), with quantification of percentage of NV area (mean ± SD; *n* = 3/group; unpaired Student’s *t*-test). Scale bars: 50 μm. (**D**) ERG measurements. Dark-adapted 0.01 ERG, Dark-adapted 3.0 ERG, Dark-adapted 3.0 Ops, and Light-adapted 3.0 ERG responses were recorded and compared among the normal control, IgG control and anti-Spp1 antibody-injected groups. Wave graphs (left) indicating the average wave forms from each group, with corresponding histograms of the wave amplitude quantifications showing in bar graphs (right). (mean ± SD; *n* = 3/group; one-way ANOVA). * *p* < 0.05, ** *p* < 0.01, *** *p* < 0.0001, **** *p* < 0.0001.

**Table 1 jpm-13-00146-t001:** Primer sequences used in the study.

**β-actin forward**	GGCTGTATTCCCCTCCATCG
**β-actin reverse**	CCAGTTGGTAACAATGCCATGT
**Kit forward**	GAGTTCCATAGACTCCAGCGTC
**Kit reverse**	AATGAGCAGCGGCGTGAACAGA

## Data Availability

Raw data have been deposited in the National Center for Biotechnology Information’s (NCBI) Sequence Read Archive (SRA) database (PRJNA864092, https://www.ncbi.nlm.nih.gov/sra (accessed on 1 August 2022)) and Genome Sequence Archive of the BIG Data Center (CRA008550, http://bigd.big.ac.cn/gsa (accessed on 19 October 2022)). Bioinformatics analysis was performed using Omicsmart (http://www.omicsmart.com (accessed on 24 September 2020 and 7 January 2022)).

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
