# Peer review of "Microglia-Derived Spp1 Promotes Pathological Retinal Neovascularization via Activating Endothelial Kit/Akt/mTOR Signaling"

_jpm, 2023, doi:10.3390/jpm13010146_

Round 1
Reviewer 1 Report
The authors demonstrated that microglia-derived Spp1 (AKA Osteopontin) promoted proliferation and migration of vascular endothelial cells through activation of Kit/Akt/mTOR signaling, associated with pathological retinal neovascularization using oxygen-induced retinopathy (OIR) model and cell lines of microglia and vascular endothelial cells. The topic is important. The study was well designed. The interpretation of the results is clear. There may be only minor points to be revised.
(page 1, line 12 in the abstract; page 16, line 5) The word, ‘improve’ may be replaced with ‘promote’, ‘enhance’, or ‘facilitate’.
(page 1, the last line in the abstract) Spp1 is a potential ‘target’ to treat ischemic ocular diseases.
(page 2, line 9) The phase, ‘rescue RNV’ may be inadequate and replaced with ‘suppress RNV’.
Author Response
Dear Reviewer:
Thank you very much for reviewing our manuscript entitled “Microglia-derived Spp1 promotes pathological retinal neovascularization via activating endothelial Kit/Akt/mTOR signaling (Manuscript ID: jpm-2073583). We have read your comments carefully and modified the manuscript accordingly. All the revisions are marked up using the “Track Changes” function and are highlighted. A point-to-point response is attached below.

Reviewer 2 Report
This study is an experimental study. In microglia of the OIR model, spp1 is the most up-regulated gene. It is said that Spp1 stimulates endothelium through Akt/mTOR to induce proliferation. This has been proven through in vitro and in vivo experiments. It's well designed and seems to produce good results. However, please correct the following items.
In Figure 2, spp1 is increased in the OIR model, but it is unknown that it is increased in microglia. In D, most cells do not colocalize Iba1 and spp1. If so, it means that spp1 is expressed in cells other than microglia. Further explanation or experimentation is needed here.
In Figure 6A, the injection treatment is a mouse, so draw the lens larger. In the ERG graph of D, the curve graph on the left and the bar graph on the right don't seem to match each other. Please fix. ERG requires a positive control, i.e. a normal control group.
The results seem to come out neatly except for ERG. If you can see it additionally, it would be more interesting to the readers if you present a slot of an experiment comparing it with anti-VEGF, which is well known to inhibit neovascularization of the retina.
Author Response

(The authors gave the same response as above.)

Round 2
Reviewer 2 Report
I think it would be good to accept it as it is now.
Author Response
Thanks again for your kind advices.